# Relationship Between Shoulder Rotation Strength and Upper Extremity Functional Assessments in Collegiate Baseball Players

**DOI:** 10.3390/jfmk10020120

**Published:** 2025-04-03

**Authors:** Andy Waldhelm, Jaclyn Aida, Jackson Boyd, Garrett Chandler, Matthew Demboski, Caroline Monson, Neil Schwarz

**Affiliations:** 1School of Physical Therapy, South College, 400 Goody Lane, Knoxville, TN 37922, USA; 2Department of Physical Therapy, University of South Alabama, 5121 USA Dr. North, Mobile, AL 36688, USAjjb2122@jagmail.southalabama.edu (J.B.); ghc1521@jagmail.southalabama.edu (G.C.); mjd1622@jagmail.southalabama.edu (M.D.); cjm2121@jagmail.southalabama.edu (C.M.); 3Department of Health, Kinesiology, and Sports, University of South Alabama, 171 Student Services Drive, HKS 1016, Mobile, AL 36688, USA; neilschwarz@southalabama.edu

**Keywords:** baseball, return-to-sport assessment, upper extremity injuries, overhead sports

## Abstract

**Background/Objectives**: In overhead throwing sports such as baseball and softball, upper extremity injuries are prevalent at both collegiate and high school levels. Currently, there is no universal assessment protocol to identify athletes at risk for injury or to determine their readiness to return to sport. This study aimed to investigate the relationship between shoulder internal rotation (IR) and external rotation (ER) isometric strength in a throwing position and three upper extremity functional tests among collegiate baseball players. It was hypothesized that there would not be significant correlations between shoulder IR and ER peak isometric force and the following functional assessments: the Upper Quarter Y Balance Test (YBT-UQ), seated single-arm shot put, and Closed Kinematic Chain Upper Extremity Strength Test (CKCUEST). **Methods**: Forty healthy collegiate baseball players volunteered for the study. After completing a self-guided warm-up, participants performed bilateral isometric shoulder IR and ER strength tests at 90 degrees of shoulder abduction and elbow flexion, followed by the three functional tests in random order. Relationships were analyzed using Pearson’s correlation coefficients (r), with a significance level set at *p* < 0.05. **Results**: Correlations were generally low, ranging from r = 0.001 to r = 0.551. Significant correlations were observed between the CKCUEST and dominant IR strength (r = 0.345, *p* = 0.031), dominant ER strength (r = 0.407, *p* = 0.010), and non-dominant ER strength (r = 0.551, *p* < 0.001). Additionally, a significant correlation was found between the dominant ER/IR strength ratio and the dominant arm superolateral reach on the YBT-UQ (r = −0.352, *p* = 0.026). No significant correlations were identified between isometric shoulder strength and the single-arm shot put. **Conclusions**: Most correlations were low to moderate, and only significant correlations between shoulder rotation isometric strength and one direction of the YBT-UQ and the CKCUEST were observed. Thus, shoulder rotation strength in a position similar to the cocking phase of overhead throwing may be valuable for return-to-sport criteria and injury screening for overhead athletes. However, further research is needed to validate these findings.

## 1. Introduction

In overhead throwing sports such as baseball and softball, the shoulder and elbow are particularly vulnerable to injury due to the high demands of repetitive, extreme torques and joint reaction forces. In Major League Baseball, of the 3512 injuries documented between 2010 and 2016, 49% were related to the upper limb [1]. This included 1229 shoulder injuries and 492 elbow injuries [1]. A nationwide survey of 11,134 junior high baseball players found that 28% of participants experienced shoulder or elbow pain over the course of one year [2]. It is estimated that 17% of all injuries sustained in baseball involve the shoulder, making it the most vulnerable anatomical location in the sport [1,3]. Elbow injuries occur at a lower rate of approximately 9.5%. [4] The average time lost due to a shoulder injury is 69 days, compared to 14 to 21 days for an elbow injury [3,4,5].

Compared to lower extremity return-to-sport testing, there is a lack of literature on the appropriate selection of tests for assessing whether an overhead athlete is ready to return to sport [6,7]. Most screening assessments are focused on strength and functionality. Assessment tools and procedures can be utilized to help screen for potential risk of injuries. Functional tests such as the CKCUEST, YBT-UQ, and seated single-arm shot put are also frequently incorporated into screening assessments for injury [8,9,10]. While these tests have not shown much diagnostic capability in the literature up to this point, they still provide useful information regarding dynamic stability, symmetry, muscular endurance, and power [11]. The seated medicine ball throw is particularly of use as it is one of the few functional tests with a throwing activity [9].

Testing shoulder IR and ER strength is a common method for assessing shoulder injury as part of return-to-sport criteria. It is believed that there is an association between shoulder rotation strength and arm injury; however, research has shown only a weak link between shoulder rotator strength and injury risk in baseball players [12,13]. Specifically, athletes with weaker internal and external rotation strength tend to experience more frequent arm pain after throwing [12]. However, there is a lack of evidence examining the relationship between shoulder rotator strength and commonly used functional tests, particularly in competitive baseball players [14].

The purpose of this study is to explore the relationship between shoulder internal and external isometric strength and upper extremity functional tests (return to sport) in baseball players. We hypothesize that baseball players will demonstrate low or no significant correlations between shoulder internal and external isometric strength and the YBT-UQ, single-arm shot put, and CKCUEST.

## 2. Materials and Methods

### 2.1. Study Design

This study utilized a cross-sectional, correlational research design.

### 2.2. Participants

The study participants consisted of 40 healthy male collegiate baseball players, including 36 right-handed and 4 left-handed players (Table 1). A power analysis indicated that a minimal total sample of 8 subjects was needed to conduct a correlation study with pilot data where r = 0.85. Inclusion criteria were being male, a current college student, a baseball player, and in good health. Exclusion criteria included the following: injury or lack of clearance from the sports medicine team or coaching staff, not being a baseball player, and not being enrolled as a college student.

### 2.3. Procedures

Testing took place outdoors on an artificial turf surface at the baseball team’s hitting facility. Prior to testing, the players completed a team dynamic warm-up. Participants then rotated through a series of upper extremity tests in a random order. The tests included the following: bilateral isometric IR and ER strength, bilateral single-arm shot put test, the CKCUEST, and the bilateral YBT-UQ.

Isometric shoulder IR and ER strength were assessed using handheld dynamometry. For IR testing, the participant lay supine on a plinth with the tested arm off the edge of the table, positioned at 90° shoulder abduction and ER and 90° elbow flexion with the forearm in pronation. For ER testing, the participant was positioned prone on the plinth with the arm in 90° abduction and ER and 90° elbow flexion, and the forearm fully pronated. The testing protocol was similar to one described by Chen et al. and testing in these positions has been shown to have good to excellent reliability [15].

In both tests, the dynamometer was positioned just proximally to the wrist [15]. Participants were instructed to press as hard as possible into the dynamometer while the tester applied resistance for three seconds. Each test consisted of one practice trial followed by two recorded attempts [15]. Shoulder IR strength was tested on both arms first, after which participants changed positions for ER testing, which was also performed on both sides. To minimize compensation, a second tester stabilized the participant’s shoulders to prevent activation of non-shoulder rotator muscles during the assessments.

The single-arm shot put test measures single-arm power and has good inter-session reliability when performed with athletes [16]. The protocol used was described by Degot et al. [16]. To perform the test, the participants stood with their backs and heads against a wall, with their knees extended in front of them [16]. In this position, they were instructed to throw a 4 kg ball at shoulder level as far as possible using one arm [16].

Compensations were not permitted during the test. These included lifting the back or head off the wall, rotating the trunk, or moving the elbow away from the trunk during the throw. If any of these compensations occurred, the throw was not recorded, and the participant was allowed to reattempt. Each participant completed one practice trial with each arm, followed by two recorded attempts with each arm. The average distance from both trials was used in the data analysis. The distance for each throw was measured from the wall to the point where the ball first contacted the ground and recorded in feet.

The CKCUEST assesses upper extremity stability since the test requires a shift in weightbearing from one upper extremity to the other, agility with a change in both speed and direction and muscle endurance as the duration of the test is fifteen seconds [17]. This study followed a standard protocol described by Tucci et al. [18] and the test with excellent reliability. The participant began in a push-up position with their hands placed on pieces of tape positioned 36 inches (91.4 cm) apart and was instructed to alternate tapping each piece of tape by reaching across with the opposite arm [18]. The test was timed for 15 s, during which the participant aimed to tap the tape as many times as possible without failure or compensatory movements. The test was performed three times, with 45 s of rest between each trial. The average number of taps from the three trials was used for analysis.

The YBT-UQ is a functional assessment designed to challenge both mobility and stability. It tests the stability of the stance limb while simultaneously requiring mobility of the thorax and reach limb [19]. During each reach, the athlete must engage their scapular stability, thoracic rotation, and core stability, as they are encouraged to reach as far as possible without losing balance [19]. By reaching outside a narrow base of support, the athlete must utilize balance, proprioception, strength, and a greater range of motion [20]. The YBT-UQ has demonstrated good to excellent reliability [21].

Before performing the test, participants were shown a demonstration and given verbal instructions. They were instructed to maintain a push-up position, with their feet no more than twelve inches apart, and to maximally reach in three directions—medial, superolateral, and inferolateral—using their free hand. At all times, the stabilizing hand was to remain in a standardized location on the platform. The distance reached in each direction was recorded. Each participant was allowed one practice trial per hand, followed by two consecutive testing trials, with the average score from both trials used for analysis.

Violations during the test include the thumb of the stabilizing hand crossing the red tape, feet positioned too far apart, failing to maintain at least one digit on the red tape of the sliding block, a complete loss of balance, or using force to push the block rather than sliding it. If a violation occurred, no score was recorded, and the participant was allowed to make another attempt.

### 2.4. Statistical Analysis

Data were analyzed using SPSS Version 26 (IBM, Armonk, NY, USA). Pearson correlation coefficients were used to estimate the relationship between shoulder internal and external isometric strength and each functional assessment with a significance level of 0.05.

## 3. Results

Descriptive statistics for each variable can be viewed in Table 1. Correlations ranged from r = 0.001 to r = 0.551 between isometric shoulder IR and ER strength measurements and the functional tests (Table 2). Significant correlations included relationships between the CKCUEST and dominant IR and dominant and non-dominant ER strength, with non-dominant IR strength trending toward significance. No significant correlations between shoulder rotation strength and the Y-balance measurements or the single-arm shot put tests were observed.

## 4. Discussion

The aim of this study was to investigate the relationship between isometric shoulder IR and ER strength, tested at 90˚ of shoulder abduction/elbow flexion, and performance on upper extremity functional tests in baseball players. The hypotheses that shoulder internal and external isometric strength would have low correlations with performance on the YBT-UQ, the single-arm shot put, and the CKCUEST were partially supported with low-to-moderate correlations, and three significant correlations were observed between shoulder IR and ER strength and performance on the CKCUEST. No significant correlations were found between shoulder IR and ER strength and performance on the YBT-UQ and single-arm shot put.

The average standardized force production (%BW) for IR and ER in both the dominant (D) and non-dominant (ND) arms was significantly lower than the values reported in previous studies [22,23,24]. In this study, the average IR strength was 22.4% (D) and 17.2% (ND), while the average ER strength was 12.7% (D) and 12.3% (ND). In contrast, Schilling et al. reported average isometric IR strength in baseball players to be 24.86% (D) and 23.75% (ND), with ER strength of 22.31% (D) and 21.39% (ND) [10]. 

Significant differences exist between the testing procedures in this study and those of previous investigations. The positions of shoulder abduction and rotation have a substantial impact on ER and IR force production, which should be considered when comparing our results to those of previous studies [10,24,25]. Schilling et al. had participants positioned with their testing arm in 90° abduction and 90° elbow flexion (90/90) with the forearm pronated, but had the glenohumeral joint in neutral rotation, not at 90˚ of external rotation [10]. To explain these discrepancies further, McWilliams et al. concluded that as the shoulder approaches neutral rotation, the ER/IR ratio gradually increases, with 90° rotation yielding the lowest ratio [24]. In their study establishing normative data for isometric rotator cuff strength in overhead athletes using a handheld dynamometer, Cools et al. tested isometric IR and ER in 90° shoulder abduction and neutral rotation, as well as a 90/90 position with 90° shoulder abduction and 90° external rotation, similar to the testing positions used in this study [25]. In the neutral rotation position, they found ER/IR ratios of 0.89 for the dominant arm and 0.93 for the non-dominant arm, while in the 90/90 position, the ratios were 0.59 for the dominant arm and 0.69 for the non-dominant arm, which are similar to the ratios observed in this study [25]. Additionally, a study by Ellenbrick et al., examining isokinetic shoulder rotation strength in professional baseball players, suggested that the selective concentric strength development of internal rotators in throwing athletes occurs without concurrent increases in external rotators [26]. 

The glenohumeral joint is a highly mobile joint, and this mobility is particularly crucial for overhead throwing athletes. The shoulder rotator muscles play a key role in the dynamic stability of the GH joint, as well as in the acceleration and deceleration of the arm during throwing, helping maximize performance while reducing the risk of injury [26]. The 90/90 position was used to assess isometric shoulder rotation strength, as it closely replicates the functional position of overhead throwing. However, this position may limit participants’ ability to generate optimal isometric internal and external rotation forces at the glenohumeral joint.

No significant correlations were found between the single-arm shot put test and isometric strength of the IR and ER, which may be attributed to the nature of the shot put movement. This test primarily involves a pushing motion that activates muscles such as the pectoralis major, triceps, and deltoid [16], rather than the rotator cuff muscles. Furthermore, the rotator cuff muscles mainly serve as stabilizers of the shoulder, whereas the shot put test is designed to assess power [9]. In contrast, when examining how baseball players threw the medicine ball, it was observed that the players released the ball at approximately 48˚ from the horizontal plane, which aligns with previous studies [9]. In baseball, the range of motion during the throwing action is greater than that in the shot put, likely leading to increased recruitment of the internal rotators.

Significant correlations were found between the CKCUEST and both dominant and non-dominant ER strength, as well as dominant IR strength. This may be attributed to the CKCUEST’s focus on shoulder joint stability, with the rotator cuff muscles playing a key role in maintaining the dynamic stability of the joint. Lee et al. also found significant correlations between the CKCUEST and shoulder rotational strength, which aligns with our findings; however, their correlations were much larger, likely due to the use of isokinetic testing rather than isometric testing [27]. The lower correlation values in our study may also be attributed to the average number of repetitions performed on the CKCUEST. In our study, participants performed an average of 25.4 repetitions, whereas Roush et al., who also studied collegiate male baseball players, had an average of 30.41 repetitions [28]. Although participants were provided adequate rest, the lower scores in our study may be due to fatigue, as the baseball players completed a short practice session prior to testing.

With correlations similar to or lower than the current study, Decleve et al. found that a modified CKCUEST had weak associations with isometric shoulder rotation strength [29]. Using a modified testing position, there was a smaller distance between hand positions, with hands placed at a distance equal to the inter-acromial distance, which may have required less stability and led to the lower correlations [29]. Furthermore, population differences may be an additional explanation for the lower correlations, as Decleve et al. used younger athletes, both male and female, from non-overhead throwing sports, basketball, and volleyball [29].

There were no significant correlations found between shoulder rotation strength and the YBT-UQ. These findings align with a study by Silva Barros et al., who investigated the correlation between isometric shoulder rotation strength and other physical assessments and the YBT-UQ in volleyball and handball athletes, using the same testing protocols as the current study [30]. They concluded that the YBT-UQ requires more core stability/strength than isometric shoulder rotation strength, as higher and more significant correlations were found between the YBT-UQ and trunk strength tests [30].

This study had a few limitations. Due to time constraints, the researchers were only able to conduct two trials per assessment. Additionally, data collection took place after the participants had completed a short practice session, which could have introduced fatigue and affected their performance, as testing was conducted outdoors on a hot afternoon. Ideally, three trials per test should be performed after a standardized warm-up protocol for a more controlled environment. The testing protocols used in this study differed slightly from those of comparable studies, making it difficult to directly compare the results with previous research. Furthermore, this study was limited to a sample from a single college baseball team, so additional research with a larger, more diverse population is needed to generalize the findings.

In future studies, including athletes from different sports, levels, and both sexes would help establish normative values for each group. This study also included participants with previous injuries, so excluding them would allow for a focus on non-injured athletes. Additionally, investigating the validity of using isometric shoulder rotation strength at 90° of shoulder abduction could serve as an injury predictor or as a return-to-sport assessment would also be valuable, as this is a simple and accessible test to administer.

## 5. Conclusions

The CKCUEST was the only test found to have a significant, but low, correlation with isometric shoulder internal and external rotation. However, no significant correlations were found between the YBT-UQ, single-arm shot put tests, and isometric internal or external rotation strength. These findings suggest that isometric shoulder rotator strength does not play a significant role in performing the functional tests used in this study. Therefore, shoulder rotation strength should be assessed independently of the functional tests. Further investigation is needed to determine the potential of isometric shoulder rotator strength measured in the 90/90 position for use in return-to-sport assessment or injury prediction.

## Figures and Tables

**Table 1 jfmk-10-00120-t001:** Descriptive statistics (mean ± standard deviation).

Age (years)	20.2 ± 1.56
Height (cm)	13 ± 7.47
Weight (kg)	89.8 ± 12.9
Dominant internal rotation strength (% BW)	22.4 ± 6.58
Non-dominant internal rotation strength (% BW)	17.2 ± 6.05
Dominant external rotation strength (% BW)	12.7 ± 4.41
Non-dominant external rotation strength (% BW)	12.3 ± 5.04
Dominant shot put (m)	4.30 ± 0.74
Non-Dominant shot put (m)	3.86 ± 0.53
CKCUEST (repetitions)	25.4 ± 4.65
Dominant YBT-UQ Medial (cm)	89.2 ± 9.75
Dominant YBT-UQ Superolateral (cm)	67.2 ± 12.0
Dominant YBT-UQ Inferolateral (cm)	93.0 ± 12.3
Average Dominant YBT-UQ (cm)	83.1 ± 9.50
Non-dominant YBT-UQ Medial (cm)	89.5 ± 10.7
Non-dominant YBT-UQ Superolateral (cm)	66.3 ± 12.7
Non-dominant YBT-UQ Inferolateral (cm)	91.7 ± 12.4
Average Non-dominant YBT-UQ (cm)	82.5 ± 10.2

Note: BW: body weight, CKCUEST: Closed Kinetic Chain Upper Extremity Test; YBT-UQ: Upper Quarter Y-Balance Test.

**Table 2 jfmk-10-00120-t002:** Correlations between shoulder internal and external strength and functional tests.

	Dom IR str	ND IR str	Dom ER str	ND ER str
Dom Shot Put	0.001	−0.087	0.026	−0.030
ND Shot Put	0.032	0.019	0.038	−0.018
% Dom/ND shot put	−0.057	−0.182	−0.025	−0.053
CKCUEST	**0.345**	0.246	**0.407**	**0.551**
Dom YBT-UQ medial	0.097	0.214	−0.007	−0.046
Dom YBT-UQ Superolateral	0.082	0.087	−0.164	−0.204
Dom YBT-UQ Inferolateral	0.042	0.150	−0.163	−0.171
Average Dom YBT-UQ	0.086	0.175	−0.142	−0.175
ND YBT-UQ medial	0.135	0.279	0.072	−0.023
ND YBT-UQ Superolateral	0.108	0.127	−0.116	−0.169
ND YBT-UQ Inferolateral	−0.026	0.086	−0.162	−0.226
Average ND YBT-UQ	0.082	0.186	−0.089	−0.171

None: **BOLD**: significant at *p* < 0.05, Dom: dominant, ND: non-dominant, IR: internal rotation, ER: external rotation, Str: strength, CKCUEST: Closed Kinetic Chain Upper Extremity Test, YBT-UQ: Upper Quarter Y-Balance Test.

## Data Availability

The data presented in this study are available on request from the corresponding author as requested by coaches of participating athletes.

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
