# Peer review of "Relationship Between Shoulder Rotation Strength and Upper Extremity Functional Assessments in Collegiate Baseball Players"

_jfmk, 2025, doi:10.3390/jfmk10020120_

Round 1

Reviewer 1 Report

Comments and Suggestions for Authors

I thank the authors for their research, which provides valuable insights into the issue at hand. I would like to comment on the necessary corrections. 
Summary: 
The text is quite extensive and requires optimization. I propose focusing on the essential statements of the study, without the need to rewrite entire paragraphs. 
Introduction: 
The claims made in lines 50-55 and 68-69 require substantiation. Sources must be cited to support these assertions. Materials and Methods: Participants: The information to be described is provided in Table 1 (subjects' data). 
Procedures: 
Although the tests used are outlined, it remains unclear whether they adhere to standardized methodologies. If they do, please provide the relevant sources; if not, please share information regarding the validity of the tests. 
Conclusion: It is essential to summarize the contribution of this study to the current field of knowledge and its practical value. 
Sincerely.

Author Response

REVIEWER 1

Comments 1: The text is quite extensive and requires optimization. I propose focusing on the essential statements of the study, without the need to rewrite entire paragraphs. 
Response 1: Thank you for the recommendation. We have made edits to the introduction to make it more concise.

Comment 2: Introduction: The claims made in lines 50-55 and 68-69 require substantiation. Sources must be cited to support these assertions.

Response 2: We have added the following references which address this issue. References have been added and the in-text referencing numbering has been adjusted. Thank you.

Pontillo M, Sennett BJ, Bellm E. Use of an upper extremity functional testing algorithm to determine return to play readiness in collegiate football players: a case series. Int J Sports Phys Ther. 2020 Dec;15(6):1141-1150.

Gorman PP Butler RJ Plisky PJ Kiesel KB. Upper Quarter Y Balance Test: reliability and performance comparison between genders in active adults. J Strength Cond Res. 2012;26(11):3043-8. 

Schilling DT, Elazzazi AM. Shoulder strength and closed kinetic chain upper extremity stability test performance in Division III Collegiate Baseball and softball players. Int J Sports Phys Ther. 2021;16(3). doi:10.26603/001c.24244

Comment 3: Materials and Methods: Participants: The information to be described is provided in Table 1 (subjects' data). 
Response 3: Thank you, we added Table 1 to the description of the subjects

Comment 4: Procedures: Although the tests used are outlined, it remains unclear whether they adhere to standardized methodologies. If they do, please provide the relevant sources; if not, please share information regarding the validity of the tests. 

Response 4: We added statements addressing the protocols or standards we used.

Comment 5: Conclusion: It is essential to summarize the contribution of this study to the current field of knowledge and its practical value. 

Response 5: Added this sentence to summarize the contribution of this study: These findings suggest that isometric shoulder rotator strength measures different components of the functional tests used in this study and therefore, shoulder rotation strength should be assessed independently from the functional tests.

Reviewer 2 Report

Comments and Suggestions for Authors

Line 66 what "link". Please provide the strength of this relationship

Are participants males or females?

"The CKCUEST assesses upper extremity stability, agility, muscle endurance, and power" I believe this statement is ambitious and the authors need to provide a more realistic overview of this test

Power analysis is missing

RESULTS

External rotation strength values are significantly lower than those provided for example by Riemann et al. The authors explain that the shoulder was placed at 90° ER but this is not described in the procedures

"The GH joint is inherently unstable, and this instability..." this is a strong statement and should be reworded. The shoulder is very mobile and this is very positive for overhead athletes. The 90/90 position is not sufficient to place the shoulder at its extreme ER ROM. Please revise this paragraph. The ability to generate force at the selected angle depends on the force-length relationship of the External Rotators, which is a point to discuss

"These rotator cuff muscles are not typically 220 considered power or pushing muscles" but rotator cuff muscles are however important to generate torque. Please revise this sentence

"The results showed a low but significant correlation between dominant ratio ER/IR strength and dominant arm superolateral reach on the YBT-UQ" I don't see the point of correlating a ratio with a performance score. I would remove from this manuscript this type of correlations

"the 90/90 position is a reliable measure for return-to-sport or injury prediction" you have not tested these, so more caution is needed when making these claims

Author Response

REVIEWER 2

Comment 1: Line 66 what "link". Please provide the strength of this relationship

Response 1: Added to the sentence: It is believed that there is an association between shoulder rotation strength and arm injury; however, research has shown only a weak link between shoulder rotator strength and injury risk in youth baseball players.10 Specifically, athletes with weaker internal and external rotation strength tend to experience more frequent arm pain after throwing.10

Comment 2: Are participants males or females?

Response 2: Males, this was added to the description of the participants. Thank  you.

Response 3: "The CKCUEST assesses upper extremity stability, agility, muscle endurance, and power" I believe this statement is ambitious and the authors need to provide a more realistic overview of this test

Response 3: We have edited this sentence. The CKCUEST assesses upper extremity stability since the test requires a shift of weightbearing from one upper extremity to the other, agility with a change in both speed and direction, and muscle endurance as the duration of the test is fifteen seconds.

Comment 4: Power analysis is missing

Response 4: Good point. A power analysis based on the pilot data where r=.85

Comment 5: RESULTS: External rotation strength values are significantly lower than those provided for example by Riemann et al. The authors explain that the shoulder was placed at 90° ER but this is not described in the procedures

Response 5: Thank you, we added it to the methods section for both IR and ER isometric testing.

Comment 6: "The GH joint is inherently unstable, and this instability..." this is a strong statement and should be reworded. The shoulder is very mobile and this is very positive for overhead athletes. The 90/90 position is not sufficient to place the shoulder at its extreme ER ROM. Please revise this paragraph. The ability to generate force at the selected angle depends on the force-length relationship of the External Rotators, which is a point to discuss

Response 6: Paragraph edited: The glenohumeral joint is a highly mobile joint, and this mobility is particularly crucial for overhead throwing athletes. The shoulder rotator muscles play a key role in the dynamic stability of the GH joint, as well as in the acceleration and deceleration of the arm during throwing, helping to maximize performance while reducing the risk of injury.25 The 90/90 position was used to assess isometric shoulder rotation strength, as it closely replicates the functional position of overhead throwing. However, this position may limit participants' ability to generate optimal isometric internal and external rotation forces at the glenohumeral joint.

Comment 7:"These rotator cuff muscles are not typically  considered power or pushing muscles" but rotator cuff muscles are however important to generate torque. Please revise this sentence

Response 7: Good point, the sentence has been removed.

Comment 8:"The results showed a low but significant correlation between dominant ratio ER/IR strength and dominant arm superolateral reach on the YBT-UQ" I don't see the point of correlating a ratio with a performance score. I would remove from this manuscript this type of correlations

Response 8: I agree, I have removed the ratio data from the manuscript.

Comment 9: "the 90/90 position is a reliable measure for return-to-sport or injury prediction" you have not tested these, so more caution is needed when making these claims

Response 9: Sentence edited: Further investigation is needed to determine the potential of isometric shoulder rotator strength measured in the 90/90 position for use in the return-to-sport assessment or for injury prediction.

Round 2

Reviewer 1 Report

Comments and Suggestions for Authors

Thanks to the authors.
I think the corrections are appropriate.
I propose to print the article.
Sincerely.

Reviewer 2 Report

Comments and Suggestions for Authors

Good work